# DelibGAN: Coarse-to-Fine Text Generation via Adversarial Network

## Abstract

In this paper, we propose a novel adversarial learning framework, namely **Delib-GAN**, for generating high-quality sentences without supervision. Our framework consists of a coarse-to-fine generator, which contains a first-pass decoder and a second-pass decoder, and a multiple instance discriminator. And we propose two training mechanisms **DelibGAN-I** and **DelibGAN-II**. The discriminator is used to fine-tune the second-pass decoder in DelibGAN-I and further evaluate the importance of each word and tune the first-pass decoder in DelibGAN-II. We compare our models with several typical and state-of-the-art unsupervised generic text generation models on three datasets (a synthetic dataset, a descriptive text dataset and a sentimental text dataset). Both qualitative and quantitative experimental results show that our models produce more realistic samples, and DelibGAN-II performs best.

## 1 Introduction

Building a good text generative model has always been a fundamental problem in the natural language processing field. The most common generative model is Recurrent Neural Networks (RNN)(Mikolov et al., 2011), which predicts each word of a sentence conditioned on the previous word and an evolving hidden state. However, it suffers from two main drawbacks: First, RNN based models are always trained through the maximum likelihood approach, which suffers from exposure bias(Bengio et al., 2015); Second, the loss function used to train the model is at word level but the performance is typically evaluated at sentence level(Wang & Wan, 2018a).

Some recent studies have tried to solve these problems. Some approaches work indirectly by making the hidden state dynamics predictable (Professor Forcing (Goyal et al., 2016)) or by randomly adjusting the sampling words during training time (Scheduled Sampling (Bengio et al., 2015)). But they do not directly specify the cost function on the RNN output to encourage high sample quality. Later works used Generative Adversarial Networks (GAN)(Goodfellow et al., 2014) or reinforcement learning to directly affect the decoding sequence, such as Gumbel-softmax distribution(Kusner & Hernández-Lobato, 2016), SeqGAN(Yu et al., 2017), MaskGAN(Fedus et al., 2018), etc. However, they all adopt an one-pass forward decoding , which can only see words that have been decoded. Recently, coarse-to-fine generator with multiple decoders has achieved great success on some tasks. But they either do not define the meaning of the sketch(Xia et al., 2017), or are limited to specific tasks such as logical form parsing, code generation, SQL query generation, etc(Lapata & Dong, 2018). And they all are supervised learning methods.

Inspired by Deliberation Networks(Xia et al., 2017), we propose a novel adversarial learning framework, namely **DelibGAN**, for generating high-quality sentences without supervision. Our framework consists of a coarse-to-fine generator and a multiple instance discriminator, and the coarse-to-fine generator contains a first-pass decoder and a second-pass decoder. Further, we propose two training mechanisms, named **DelibGAN-I** and **DelibGAN-II**. The former one uses the sentence-level penalty predicted by the discriminator to fine-tune the second-pass decoder. And the latter one imposes the influence of word importance on the first-pass decoder, with the purpose of decoding important words as much as possible in the first pass, and thus better guiding the decoding of the second-pass decoder. The word importance is obtained through the multiple instance learning mechanism of the discriminator, which is used to directly tune the first-pass decoder and later guide the second-pass decoding. In short, for DelibGAN-I, the discriminator is used to fine-tune the second-

pass decoder $\mathcal{G}_2$, while for DelibGAN-II, the discriminator is further used to evaluate the importance of each word and tune $\mathcal{G}_1$.

Our motivation is two-fold: (1) We use a coarse-to-fine generator to generate important words (i.e., sketch) as much as possible during the first decoding and utilize the global information (i.e., the sentence decoded in the first pass) at the time of final decoding; (2) The discriminator can evaluate the sentence-level quality to guide the final generation via policy gradients, moreover, it can provide word-level importance, which can be used to adjust the sketches in the first-pass generation to make sense.

We compare our models with several typical and state-of-the-art unsupervised generic text generation models, including RNNLM(Mikolov et al., 2011), SeqGAN(Yu et al., 2017), SentiGAN(Wang & Wan, 2018a), MaskGAN(Fedus et al., 2018). Experimental results on three datasets (a synthetic dataset, a descriptive text dataset and a sentimental text dataset) show that our models can produce more realistic samples. The major contributions of this paper are summarized as follows:

- We propose a novel adversarial learning framework - DelibGAN, which consists of a coarse-to-fine generator and a multiple instance discriminator, to generate high-quality sentences without supervision.
- We propose two training mechanisms, DelibGAN-I and DelibGAN-II, to make the generator generate higher-quality sentences.
- Both qualitative and quantitative results show that DelibGAN-I and DelibGAN-II can produce more realistic samples, and DelibGAN-II performs best.

## 2 FRAMEWORK

### 2.1 ARCHITECTURE

The architecture of our proposed DelibGAN is show in Figure1. The entire model consists of two modules: a coarse-to-fine generator and a multiple instance discriminator. The coarse-to-fine generator has a first-pass decoder $\mathcal{G}_1$ and a second-pass decoder $\mathcal{G}_2$. The multiple instance discriminator $\mathcal{D}$ can evaluate both a whole sentence and each individual word in the sentence by using the multiple instance learning mechanism, and thus the two decoders can be influenced by the discriminator $\mathcal{D}$ in different ways. In our first training mechanism DelibGAN-I, the sentence-level penalty predicted by the discriminator is used to fine-tune the second-pass decoder $\mathcal{G}_2$. In our second training mechanism DelibGAN-II, in addition to the fine-tuning of $\mathcal{G}_2$, the word-level importance is predicted and used to tune $\mathcal{G}_1$, to make $\mathcal{G}_1$ decode important words as much as possible, and thus better guide the decoding of $\mathcal{G}_2$.

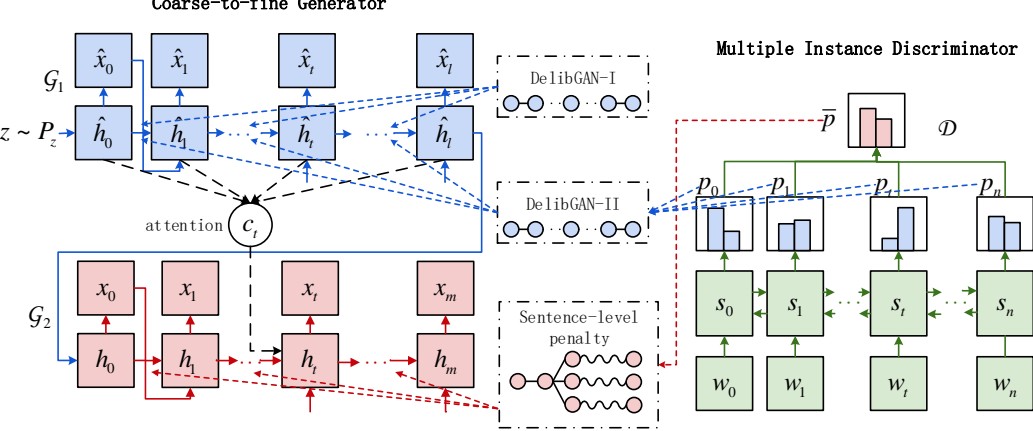

Figure 1: The architecture of DelibGAN. Two training mechanisms DelibGAN-I and DelibGAN-II are detailed in section 2.4.

## 2.2 COARSE-TO-FINE GENERATOR

Inspired by Deliberation Networks(Xia et al., 2017), we propose a coarse-to-fine generator with the following two purposes: The first is to enable the generator to utilize the global information (i.e., the sentence decoded in the first pass) at the time of final decoding, rather than only the sequence that has been decoded; The second is to force the generator to generate important words (i.e., sketch) as much as possible during the first-pass decoding, so as to provide more useful clues for decoding in the second pass.

Our coarse-to-fine generator contains a first-pass decoder $\mathcal{G}_1$ and a second-pass decoder $\mathcal{G}_2$. Both decoders are RNN and can be easily replaced with its variants such as LSTM and GRU (In this work, we use LSTM). The prior input noise z sampled from a distribution $P_z$ (e.g., a normal distribution) is used to initialize the input of $\mathcal{G}_1$, then $\mathcal{G}_1$ will generate a first-pass sequence $\hat{x} = \{\hat{x_1}, \hat{x_2} \cdots \hat{x_l}\}$, and a series of hidden states $\hat{H} = \{\hat{h_1}, \hat{h_2} \cdots \hat{h_l}\}$, where $l$ is the length of the generated sequence. At step $t$, $\hat{h}_t$ is calculated as $\hat{h}_t = \text{RNN}(f(\hat{x}_{t-1}), \hat{h}_{t-1})$, where $f$ is a word embedding representation function. Then $\hat{h}_t$ is fed into a softmax layer, and $\hat{x}_t$ is sampled out from the obtained multinomial distribution. Finally, the first-pass output sequence $\hat{x}$ is generated by $\mathcal{G}_1$.

In the second-pass decoding, we take the hidden state of the last moment of $\mathcal{G}_1$ as the initial hidden state of $\mathcal{G}_2$. And we use an attention model in $\mathcal{G}_2$, Specifically, at step $t$, the attention model in $\mathcal{G}_2$ first generates a context $c_t$ defined as follows:

$$c_t = \sum_{i=1}^{l} a_i \hat{h_i}; a_i \propto \exp\left(v_a^T \tanh(\boldsymbol{W}_{att} \hat{h_i} + \boldsymbol{U}_{att} \boldsymbol{h}_{t-1})\right) \forall i \in [1, l]; \sum_{i=1}^{l} a_i = 1. \quad (1)$$

After receiving $c_t$, we calculate the hidden state $h_t$ as $h_t = \text{RNN}([f(x_{t-1}); c_t], h_{t-1})$. Then $x_t$ is sampled in the same way of sampling $\hat{x}_t$, and the second-pass output sequence $\boldsymbol{x}$ is generated and its length is $m$.

## 2.3 MULTIPLE INSTANCE DISCRIMINATOR

Like Yu et al. (2017) and Wang & Wan (2018a), in order to evaluate the performance of a generator at the sentence-level, we use a discriminator to calculate rewards. Further, we want to get the importance of each word in the sentence to adjust the generator to decode the important words as much as possible in the first-pass decoding. Thus we adopt a multi-instance learning (MIL) mechanism in our discriminator. MIL deals with problems where labels are associated with groups of instances or bags (sentences in our case), while instance labels (word-level importance) are unobserved (Keeler & Rumelhart, 1991; Zhang et al., 2002; Wang & Wan, 2018b).

Our multiple instance discriminator $\mathcal{D}$ uses a structure similar to a bidirectional RNN. Supposing the input to $\mathcal{D}$ is a sentence $\boldsymbol{w} = \{w_1, w_2, \cdots, w_n\}$, where $n$ is the length of $\boldsymbol{w}$. We apply a bidirectional LSTM on the sentence, and for each word $w_t(t \in n)$, we concatenate its corresponding forward and backward hidden state vectors:

$$\overrightarrow{s_t} = \overrightarrow{\text{LSTM}}(w_t); \overleftarrow{s_t} = \overleftarrow{\text{LSTM}}(w_t); s_t = \overrightarrow{s_t} || \overleftarrow{s_t}, \quad (2)$$

where $\overrightarrow{\text{LSTM}}(w_t)$ and $\overleftarrow{\text{LSTM}}(w_t)$ represent the forward and backward LSTM hidden state vectors for $w_t$. Then we use a softmax classifier to get the word-level probability distribution over target labels (i.e., $real$ and $fake$).

$$\boldsymbol{p}_t = \text{softmax}(\tanh(\boldsymbol{W}_d \boldsymbol{s}_t + \boldsymbol{b}_d), \text{ where } \boldsymbol{p}_t = [p_{real}^{(t)}, p_{fake}^{(t)}]; p_{real}^{(t)} + p_{fake}^{(t)} = 1. \quad (3)$$

$p_{real}^{(t)}$ and $p_{fake}^{(t)}$ represent the probability that the $t$-th word is predicted to be real and fake, respectively; $\boldsymbol{W}_d$ and $\boldsymbol{b}_d$ are parameters, shared across all words. Finally, we use the average of the word-level distribution as the sentence-level distribution over the target labels:

$$\overline{p_i} = \frac{1}{n} \sum_{t=1}^{n} p_i^{(t)}, \forall i \in \{\text{real}, \text{fake}\}; \overline{\boldsymbol{p}} = [\overline{p}_{real}, \overline{p}_{fake}]. \quad (4)$$

## 2.4 TRAINING

Here we introduce the adversarial training of the generator and the discriminator in detail. In this paper, we formalize the text generation problem as a sequential decision making process (Bachman & Precup, 2015), which solves the problem that the gradient cannot pass back to the generative model when the output is discrete. Moreover, we use policy gradients to update parameters of the generator.

We propose two loss functions for the generator, named DelibGAN-I and DelibGAN-II. DelibGAN-I just forces the first-pass decoder $\mathcal{G}_1$ to maintain the same decoding result, while DelibGAN-II utilizes the result of the discriminator at the word level, forcing $\mathcal{G}_1$ to decode important words (evaluated by the discriminator) as much as possible. But for the second-pass decoder $\mathcal{G}_2$, both of them use the penalty-based objective function (Wang & Wan, 2018a) to update the parameters $\theta_{\mathcal{G}_2}$ of $\mathcal{G}_2$. Specifically, at each timestep $t$, the existing sequence generated by $\mathcal{G}_2$ is $\boldsymbol{x}_{1:t-1} = \{x_1, ..., x_{t-1}\}$, and the next token $x_t$ selected in the next step is an action sampling from the policy $\mathcal{G}_2(x_t|\boldsymbol{x}_{1:t-1}; \theta_{\mathcal{G}_2})$, and we use the Monte Carlo rollouts method (Dai et al., 2017; Yu et al., 2017) to simulate intermediate penalty $R_{\theta_{\mathcal{G}_2}, \theta_{\mathcal{D}}}(x_t)$, where $\theta_{\mathcal{D}}$ is the parameters of the discriminator $\mathcal{D}$. Then, the total penalty of $\mathcal{G}_2$ can be computed by:

$$
\begin{aligned}
\mathbb{E}_{\mathrm{x}\sim\mathcal{G}_2}(\boldsymbol{x}) &= \sum_{t=1}^{m}[(R_{\theta_{\mathcal{G}_2}, \theta_{\mathcal{D}}}(x_t) - b)\mathcal{G}_2(x_t|\boldsymbol{x}_{1:t-1}; \theta_{\mathcal{G}_2})] \\
&= \sum_{t=1}^{m}[(\frac{1}{q}(\sum_{j=1}^{q}(1 - \mathcal{D}(x_t|\boldsymbol{x}_{1:t-1}^{(j)}; \theta_{\mathcal{D}}))) - b)\mathcal{G}_2(x_t|\boldsymbol{x}_{1:t-1}; \theta_{\mathcal{G}_2})],
\end{aligned}
\tag{5}
$$

where $b$ is the bias, $\boldsymbol{x}_{1:t-1}^{(j)}$ is the $q$-time Monte Carlo search sampled based on the roll-out policy $\mathcal{G}_2(x_t|\boldsymbol{x}_{1:t-1}; \theta_{\mathcal{G}_2})$, and $\mathcal{D}(x_t|\boldsymbol{x}_{1:t-1}^{(j)}; \theta_{\mathcal{D}})$ is the sentence probability $\overline{p}_{real}$ given by the discriminator for sentence $\boldsymbol{x}_{1:t-1}^{(j)} \| x_t$.

**DelibGAN-I:** For DelibGAN-I, we just force the first-pass decoder $\mathcal{G}_1$ to maintain the same decoding result. Specifically, after we use $(\mathcal{G}_1, \mathcal{G}_2)$ to sample a pair of sentences $(\hat{\boldsymbol{x}}, \boldsymbol{x})$, we first use the discriminator to calculate the sentence-level penalties. Then in the process of using penalties to update the generator, we force the first-pass decoder $\mathcal{G}_1$ to decode the same result, which is $\hat{\boldsymbol{x}}$. The loss of generator $\mathcal{G}_1$ (the parameters are denoted as $\theta_{\mathcal{G}_1}$) is defined as follows:

$$
\mathbb{E}_{\mathrm{x}\sim\mathcal{G}_1}^{(I)}(\hat{\boldsymbol{x}}) = \frac{1}{l}\sum_{t=1}^{l} -\log(\mathcal{G}_1(\hat{x}_t|\hat{\boldsymbol{x}}_{1:t-1}; \theta_{\mathcal{G}_1}))
\tag{6}
$$

we propose it with the purpose of keeping the results of $\mathcal{G}_1$ as constant as possible, and just fine-tuning $\mathcal{G}_2$ based on the sentence-level penalty mentioned above. Finally, the objective of the generator is to minimize:

$$
\begin{aligned}
J_{\mathcal{G}_1, \mathcal{G}_2}^{(I)}(\theta_{\mathcal{G}_1}, \theta_{\mathcal{G}_2}) &= \mathbb{E}_{\mathrm{x}\sim\mathcal{G}_1}^{(I)}(\hat{\boldsymbol{x}}) + \mathbb{E}_{\mathrm{x}\sim\mathcal{G}_2}(\boldsymbol{x}) = \frac{1}{l}\sum_{t=1}^{l} -\log(\mathcal{G}_1(\hat{x}_t|\hat{\boldsymbol{x}}_{1:t-1}; \theta_{\mathcal{G}_1})) \\
&+ \sum_{t=1}^{m}[(\frac{1}{q}(\sum_{j=1}^{q}(1 - \mathcal{D}(x_t|\boldsymbol{x}_{1:t-1}^{(j)}; \theta_{\mathcal{D}}))) - b)\mathcal{G}_2(x_t|\boldsymbol{x}_{1:t-1}; \theta_{\mathcal{G}_2})].
\end{aligned}
\tag{7}
$$

**DelibGAN-II:** For DelibGAN-II, we want to impose direct influence to $\mathcal{G}_1$, forcing it to decode important words as much as possible in the first-pass decoding, to better guide the decoding of $\mathcal{G}_2$. We tried different ways of influencing, including using sentence-level penalties like Eq5, but finally we introduced and leveraged a multi-instance learning mechanism in the discriminator. This is because it can provide the importance of words in a sentence, and we add the word-level importance to the loss as weights. So the loss of generator $\mathcal{G}_1$ is changed as follows:

$$
\mathbb{E}_{\mathrm{x}\sim\mathcal{G}_1}^{(II)}(\hat{\boldsymbol{x}}) = \frac{1}{l}\sum_{t=1}^{l} -p_{real}^{(t)}\log(\mathcal{G}_1(\hat{x}_t|\hat{\boldsymbol{x}}_{1:t-1}; \theta_{\mathcal{G}_1}))
\tag{8}
$$

where $p_{real}^{(t)}$ is the probability that the discriminator predicts $\hat{x}_t$ as real. Therefore, the objective of the generator is to minimize the total loss:

$$J_{\mathcal{G}_1,\mathcal{G}_2}^{(II)}(\theta_{\mathcal{G}_1}, \theta_{\mathcal{G}_2}) = \mathbb{E}_{x \sim \mathcal{G}_1}^{(II)}(\hat{\boldsymbol{x}}) + \mathbb{E}_{x \sim \mathcal{G}_2}(\boldsymbol{x}) = \frac{1}{l} \sum_{t=1}^{l} -p_{real}(\hat{x}) \log(\mathcal{G}_1(\hat{x}_t | \hat{\boldsymbol{x}}_{1:t-1}; \theta_{\mathcal{G}_1}))$$
$$+ \sum_{t=1}^{m} [(\frac{1}{q}(\sum_{j=1}^{q}(1 - \mathcal{D}(x_t | \boldsymbol{x}_{1:t-1}^{(j)}; \theta_{\mathcal{D}}))) - b)\mathcal{G}_2(x_t | \boldsymbol{x}_{1:t-1}; \theta_{\mathcal{G}_2})]. \tag{9}$$

For discriminator $\mathcal{D}$, the goal is to distinguish between real text and generated text as much as possible. It is worth noting that we only have the target label for the sentence, but do not have labels on words. The objective function of the discriminator is to minimize:

$$J_{\mathcal{D}}(\theta_{\mathcal{D}}) = -\mathbb{E}_{x \sim \mathcal{G}} \log \overline{p}_{fake} - \mathbb{E}_{x \sim \mathcal{S}} \log \overline{p}_{real}, \tag{10}$$

where $\mathcal{S}$ is real sentences in the corpus. We perform the adversarial training of the generator and the discriminator, and train them alternately, as shown in Algorithm 1.

---

**Algorithm 1** The adversarial training process in DelibGAN

---

**Input:** Input noise, $z$; Generator, $(\mathcal{G}_1, \mathcal{G}_2)$; Discriminator, $\mathcal{D}$; Real text dataset, $\mathcal{S}$;
**Output:** Well trained generator, $(\mathcal{G}_1, \mathcal{G}_2)$;
1: Initialize $(\mathcal{G}_1, \mathcal{G}_2)$,$\mathcal{D}$ with random weights;
2: Pre-train $(\mathcal{G}_1, \mathcal{G}_2)$ using MLE on $\mathcal{S}$;
3: **repeat**
4:     **for** d-steps **do**
5:         Generate fake texts $(\{\hat{\boldsymbol{x}}\}, \{\boldsymbol{x}\})$ using $(\mathcal{G}_1, \mathcal{G}_2)$;
6:         Update $\mathcal{D}$ using $(\{\boldsymbol{x}\}, \mathcal{S})$ by minimizing Eq 10;
7:     **end for**
8:     **for** g-steps **do**
9:         Generate fake texts $(\{\hat{\boldsymbol{x}}\}, \{\boldsymbol{x}\})$ using $(\mathcal{G}_1, \mathcal{G}_2)$;
10:        Calculate penalty $\mathbb{E}_{x \sim \mathcal{G}_2}(\boldsymbol{x})$ by Eq (5) ;
11:        For DelibGAN-I, Update $(\mathcal{G}_1, \mathcal{G}_2)$ by minimizing Eq (7);
12:        For DelibGAN-II, Update $(\mathcal{G}_1, \mathcal{G}_2)$ by minimizing Eq (9);
13:     **end for**
14: **until** Model converges
15: **return** ;

---

## 3 EXPERIMENTS

### 3.1 SETUP

In this study, we hope that our unsupervised models can generate sentences with higher quality. Without loss of generality, we evaluate our model [1] on three benchmark datasets: a synthetic dataset, a descriptive text dataset and a sentimental text dataset. We compare with several typical and state-of-the-art unsupervised generic text generation models, including RNNLM(Mikolov et al., 2011), SeqGAN(Yu et al., 2017), SentiGAN(Wang & Wan, 2018a), MaskGAN(Fedus et al., 2018). It is worth noting that pre-training was used for all selected baselines.

After training models, we let them generate 1k sentences each. Measuring the quality of generated sentences has always been a difficult problem. For synthetic data, we have an oracle model to measure the negative log-likelihood (NLL) scores. But for real texts, we use automatic quantitative indicators (e.g., Fluency, Novelty and Diversity) and human evaluation methods (e.g., Grammaticality, Topicality and Overall) to evaluate the quality of generated sentences.

For automatic quantitative indicators, same as Wang & Wan (2018a), we use a language modeling toolkit - SRILM (Stolcke, 2002) to test the fluency of generated sentences, which calculates the perplexity of generated sentences using the language model trained on respective corpus. In addition,

---

[1]We will release the source codes of our methods upon the acceptance of this paper.

Table 1: The performance comparison on the synthetic data in terms of the NLL scores.

| Methods | MLE | SeqGAN | SentiGAN | DelibMLE | DelibGAN-I | DelibGAN-II |
|---------|-----|--------|----------|----------|------------|-------------|
| NLL | 9.038 | 8.736 | 6.924 | 8.604 | **5.594** | **5.250** |

given the generated sentence set $\mathbb{A}$ and its training corpus set $\mathbb{B}$, the novelty $Novelty(\mathbb{A}|\mathbb{B})$ and the diversity $Diversity(\mathbb{A})$ are defined as follows:

$$Novelty(\mathbb{A}|\mathbb{B}) = \frac{1}{|\mathbb{A}|}\sum_{i\in\mathbb{A}}(1 - \max\{\varphi(i,j); \forall j \in \mathbb{B}\}), \tag{11}$$

$$Diversity(\mathbb{A}) = \frac{1}{|\mathbb{A}|}\sum_{i\in\mathbb{A}}(1 - \max\{\varphi(i,j); \forall j \in \mathbb{A}\backslash\{i\}\}), \tag{12}$$

$\varphi$ is the Jaccard similarity function. It is worth noting that in terms of the fluency indicator, a smaller value is better, but the novelty and diversity indicators are just the opposite.

For human evaluation, we randomly extracted 100 sentences from the generated sentences, and then let five experts rate each sentence according to its "grammaticality", "topicality" and "overall" aspects, where "topicality" indicates whether the sentence accords with the topic/genre of the dataset (i.e., happy moment and sentimental text for the two real datasets, respectively). Scores range from 1 to 5, with 5 being the best. We finally calculate the average of the scores.

## 3.2 SIMULATION ON SYNTHETIC DATA

Here we use the synthetic dataset[2] used by Yu et al. (2017), which consists of a set of sequential tokens which can be seen as the simulated data comparing to the real-word language data. Moreover, it automatically evaluates the negative log-likelihood (NLL) scores of our generated sequences, which brings us convenience. We compare our model with various models on this dataset, as shown in Table 1 and Figure 2. It is worth noting that the DelibMLE method is just using our coarse-to-fine generator.

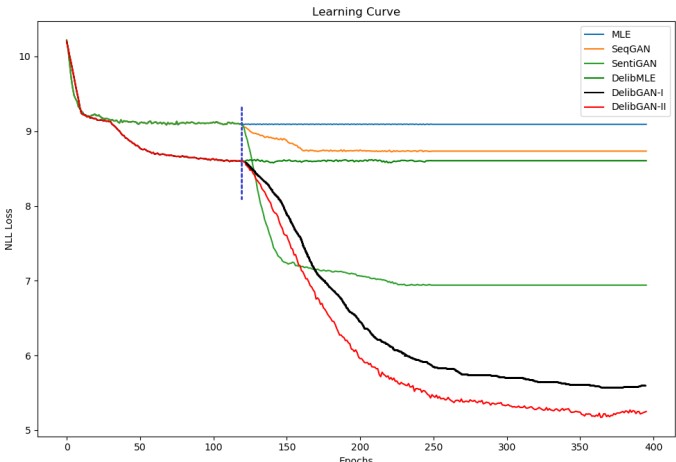

Figure 2: The illustration of learning curves. Dotted line is the end of pre-training.

From the results, we can see that: (1) Our coarse-to-fine generator (DelibMLE) performs better and even outperforms seqGAN. (2) Our proposed methods DelibGAN-I and DelibGAN-II outperform all other competitors with a large margin which means the framework we proposed is better than

---

[2]The synthetic data and the oracle model (LSTM model) are publicly available at https://github.com/LantaoYu/SeqGAN

Table 2: Comparison of automatic quantitative indicators of generated sentences on HappyDB corpus. ↓ means that the smaller the better, and ↑ is the opposite.

| Methods | SeqGAN | SentiGAN | MaskGAN | DelibMLE | DelibGAN-I | DelibGAN-II |
|---|---|---|---|---|---|---|
| Fluency(↓) | 156.212 | 130.655 | 121.273 | 138.353 | **105.593** | **98.348** |
| Novelty(↑) | 0.275 | 0.324 | 0.267 | 0.305 | **0.359** | **0.399** |
| Diversity(↑) | 0.694 | 0.732 | 0.753 | 0.723 | **0.765** | **0.767** |

Table 3: Comparison of human evaluation of generated sentences on HappyDB corpus.

| Methods | SeqGAN | SentiGAN | MaskGAN | DelibMLE | DelibGAN-I | DelibGAN-II |
|---|---|---|---|---|---|---|
| Grammaticality | 2.256 | 2.694 | 3.548 | 2.438 | **3.864** | **3.871** |
| Topicality | 2.331 | 2.652 | 3.361 | 3.237 | **3.763** | **4.026** |
| Overall | 2.384 | 2.698 | 3.525 | 2.752 | **3.644** | **3.836** |

the other models in capturing the dependency of the sequential tokens. (3) DelibGAN-II is better than DelibGAN-I, which indicates that it is useful to use the word-level information given by the discriminator during the first-pass decoding.

### 3.3 RESULTS ON DESCRIPTIVE TEXT DATA

In this section, we validate our model on the descriptive text corpus: HappyDB(Asai et al., 2018). HappyDB is a corpus of 100,000+ crowd-sourced happy moments. Its content is mainly about events that make people happy in the past, and 83% of corpora contain only one sentence. We only use those instances that contain one sentence, and we get a total of 83711 happy moments. Table 2 shows the results on the automatic quantitative indicators, and we can see that DelibGAN-I and DelibGAN-II perform exceptionally well, with the ability of keeping the fluency of sentences, generating sentences different from that in the training corpus and generating a variety of sentences.

Further, we use human evaluation to evaluate generated sentences. Table 3 shows the results, and we can see DelibGAN-I and DelibGAN-II outperform other models, especially in topicality.

### 3.4 RESULTS ON SENTIMENTAL TEXT DATA

Here we validate our model on the sentimental text corpus: Stanford Sentiment Treebank (SSTB, Socher et al. (2013)), which contains 9613 sentimental reviews of movies. Tables 4 and 5 show the results of automatic quantitative indicators and human evaluation, respectively. Both results show that our models DelibGAN-I and DelibGAN-II produce more realistic samples of sentimental texts than other models, and DelibGAN-II performs best.

#### 3.4.1 CASE STUDY

In Table 6, we show example sentences generated by DelibMLE, DelibGAN-I and DelibGAN-II. From the examples, we can see that: (1) In DelibMLE, there is no significant correlation between the first-pass decoding result $\mathcal{G}_1$ and the second-pass decoding result $\mathcal{G}_2$. (2) Compared with DelibGAN-I, DelibGAN-II is more likely to decode important words in $\mathcal{G}_1$, and those words behave like keywords in HappyDB and sentiment words in SSTB.

Table 4: Comparison of automatic quantitative indicators of generated sentences on SSTB corpus. ↓ means that the smaller the better, and ↑ is the opposite.

| Methods | SeqGAN | SentiGAN | MaskGAN | DelibMLE | DelibGAN-I | DelibGAN-II |
|---|---|---|---|---|---|---|
| Fluency(↓) | 98.253 | 82.554 | 83.967 | 89.365 | **75.573** | **71.365** |
| Novelty(↑) | 0.298 | 0.344 | 0.342 | 0.348 | **0.387** | **0.393** |
| Diversity(↑) | 0.641 | 0.711 | 0.723 | 0.673 | **0.734** | **0.740** |

Table 5: Comparison of human evaluation of generated sentences on SSTB corpus.

| Methods | SeqGAN | SentiGAN | MaskGAN | DelibMLE | DelibGAN-I | DelibGAN-II |
|---|---|---|---|---|---|---|
| Grammaticality | 2.534 | 3.712 | 3.826 | 3.529 | **3.684** | **4.037** |
| Topicality | 2.164 | 3.231 | 3.567 | 3.471 | **3.973** | **4.304** |
| Overall | 2.865 | 3.284 | 3.483 | 3.197 | **4.082** | **4.243** |

Table 6: Examples sentences generated by our models trained on HappyDB and SSTB.

| | | | |
|---|---|---|---|
| HappyDB Corpus | DelibMLE | $\mathcal{G}_1$ | *i got out . had a money money . .* |
| | | $\mathcal{G}_2$ | *i worked extra and well donut with really well recently .* |
| | DelibGAN-I | $\mathcal{G}_1$ | *i got a new car . . .* |
| | | $\mathcal{G}_2$ | *i purchased a new car for my family .* |
| | DelibGAN-II | $\mathcal{G}_1$ | *i **friends** to to visit a **games** . **weekend*** |
| | | $\mathcal{G}_2$ | *our friends came over to play video game last weekend .* |
| SSTB Corpus | DelibMLE | $\mathcal{G}_1$ | *my <UNK>review : deeply deeply romantic film that cynicism* |
| | | $\mathcal{G}_2$ | *fantastic summer thrill ride !* |
| | DelibGAN-I | $\mathcal{G}_1$ | *it was <UNK>.* |
| | | $\mathcal{G}_2$ | *it was a very important classic !* |
| | DelibGAN-II | $\mathcal{G}_1$ | *it **funny** . <UNK> **us** .* |
| | | $\mathcal{G}_2$ | *a funny story for the former child star in all of us* |

## 4 RELATED WORK

Unsupervised text generation models are usually based on RNN(Mikolov et al., 2011), which easily suffers from exposure bias(Bengio et al., 2015) and the inconsistency between word-level loss function and sentence-level evaluation(Wang & Wan, 2018a). Several strategies have been exploited later, including Professor Forcing (Goyal et al., 2016), Scheduled SamplingBengio et al. (2015) and MaskMLE(Fedus et al., 2018).

Generative Adversarial Networks (GAN)(Goodfellow et al., 2014) has also been applied for text generation. Due to the non-differentiable problem and more complex modes of languages, the quality of generated texts are usually not very satisfactory. Some studies apply GAN in text generation by modifying the word prediction function, such as Gumbel-softmax distribution(Kusner & Hernández-Lobato, 2016). Some other works apply reinforcement learning to solve the non-differentiable problem, including SeqGAN(Yu et al., 2017), LeakGAN(Guo et al., 2018), RankGAN(Lin et al., 2017), SentiGAN(Wang & Wan, 2018a) and MaskGAN(Fedus et al., 2018). However, they all adopt an one-pass forward decoding , which can only see words that have been decoded.

Recently, Deliberation networks(Xia et al., 2017), which has two decoders and produces better sequences by modifying sketches, has achieved good results on machine translation tasks. Later, Lapata & Dong (2018) also propose a similar structure and make meaningful settings for the sketches. But this is limited to specific tasks such as logical form parsing, code generation and SQL query generation. Moreover, these methods are supervised learning methods.

Multiple Instance Learning (MIL)(Keeler & Rumelhart, 1991) deals with problems where labels are associated with groups of instances or bags (sentences in our case), while instance labels (word-level importance in our case) are unobserved. The initial MIL makes a strong assumption that a bag is negative only if all of its instances are negative, and positive otherwise(Dietterich et al., 1997; Maron & Ratan, 1998; Zhang et al., 2002), and subsequent works relax this assumption and make it more suitable for the task at hand(Cour et al., 2011; Kotzias et al., 2015; Wang & Wan, 2018b).

## 5 CONCLUSIONS AND FUTURE WORK

In this paper, we propose a novel adversarial learning framework **DelibGAN** to generate high-quality sentences without supervision, by making use of a coarse-to-fine generator and a multiple instance discriminator. Evaluation results on various datasets verifies the efficacy of our proposed models. In future work, we will try to make the sketch more meaningful and apply our framework to more tasks.

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
