# OpenReview forum: "DelibGAN: Coarse-to-Fine Text Generation via Adversarial Network"
_ICLR.cc/2019/Conference_

### Official Review · AnonReviewer3 · 2018-11-01
**interesting idea but poorly written and experiments not well-executed**

**Rating:** 4
**Confidence:** 4

**Review:**

Contributions:

The main contribution of this paper is the proposed DelibGAN for text generation. The framework introduces a coarse-to-fine generator, which contains a first-pass decoder and a second-pass decoder. Instead of using a binary classifier, the discriminator is a multiple instance discriminator. Two different variants of DelibGAN are proposed, with experiments showing that DelibGAN-II performs the best.

Strengths:

(1) Novelty: I think this paper contains some novelty inside. Using a coarse-to-fine generator is an interesting idea. However, as I noted below, the paper is not well-executed.

Weaknesses:

(1) Presentation: This paper is easy to follow, but poorly written.

First, the paper is too repetitive. For example, the two-pass decoding process has been repeatedly mentioned too many times in the paper, such as the paragraph above Eqn. (8). Please be concise.

Second, when citing a paper, there should be a space between the word and the cited paper. For example, in the first paragraph of the introduction section, instead of "(RNN)(Mikolov et al., 2011)", it should be "(RNN) (Mikolov et al., 2011)". This should be corrected for the whole paper.

Third, in the first paragraph of the introduction section, I am not sure why (Wang & Wan, 2018a) is cited here. This is not the first paper that points out the problem. One should refer to [a], which is also not cited in the paper.

Missing reference: I also encourage the authors citing [b] since it is directly related to this work, which is about using GAN for text generation.

[a] Sequence Level Training with Recurrent Neural Networks, ICLR 2016
[b] Adversarial Feature Matching for Text Generation, ICML 2017

(2) Evaluation: My main concern lies in the experimental evaluation, with detailed comments listed below.

Questions:

(1) In Algorithm 1, there exists the pretraining process of G_1 & G_2. However, it is not clear to me how this this pretraining is implemented, since the output of G_1 is not observed, but is hidden and imagined by the model. So, what is the training signal for pretraining G_1? Can the authors provide more details? Please clarify it.

(2) In experiments, why the authors only compare with SeqGAN, SentiGAN & MaskGAN? One would naturally ask how the proposed model compare with RankGAN, TextGAN and LeakGAN. For example, what are the corresponding results of RankGAN, TextGAN & LeakGAN in Table 1 to 5? This should not be difficult to compare with, based on the availability of Texygen.

(3) Besides using the perplexity, the authors also use the novelty and diversity terms defined in (11) & (12) for evaluation. This is good. However, why not using some commonly used metrics in the literature, such as BLEU and self-BLEU? I agree these metrics are also not perfect, but it will be helpful to also report these results for benchmark purposes.

(4) Instead of using datasets like HappyDB & SSTB, it would be helpful to also report results on some benchmark datasets such as COCO captions and EMNLP News as used in the LeakGAN paper.  What are the results looking like on these two datasets?

(5) The results in Table 1 & Figure 2 is misleading. They do not necessarily mean the proposed model is better, as the NLL value only measures how the generated sentence look like a real sentence, but it does not measure the diversity of generated sentences. For example, a model that only repeatedly produces one very realistic sentence would also achieve a very high NLL score.

(6) Table 3 & 5 shows the human evaluation results. However, how this is performed is not clear at all. For example, is the human evaluation performed using AMT? How many samples are used for human evaluation? Also, how many workers has one sentence been evaluated by? Without all these details, how reliable this result is is questionable.

Minor issues:

(1) In Eqn. (3), since only two classes exist, there is no need to say "softmax", use "sigmoid" function is enough for illustration.

(2) In the line below Eqn. (5), "bias" => "baseline"

(3) In Eqn. (3), there is ")" missing.

(4) In Figure 2, there are two methods "SentiGAN" and "DelibMLE" with the same green color. Which one is which is hard to see.

(5) In the first paragraph of related work, MaskMLE => MaskGAN.

---

### Official Review · AnonReviewer2 · 2018-11-02
**Approach isn't clear, but seems to lack novelty. Writing could be better.**

**Rating:** 3
**Confidence:** 4

**Review:**

Summary:
The authors propose a novel adversarial learning framework consisting of a coarse-to-fine generator and a multiple instance discriminator to generate high-quality sentences without supervision, along with two training mechanisms to make the generator produce higher-quality sentences.

The coarse-to-fine generator is implemented as a two-stage decoder where the first stage decoder is initialized with a noise vector z and produces an initial sequence by sampling from its distribution at each step. The second stage decoder is able to attend to first stage decoder hidden states when generating its sequence and is initialized with the last hidden state of the first state decoder

Two training frameworks are defined to help the coarse-to-fine generator learn to produce coherent text. In both frameworks, a multi-instance learner is used as a discriminator and the overall score for the sequence is an average of word-level scores.

In DelibGAN-I, the first stage decoder is just trained to minimize the negative log likelihood of producing the sampled sequence. I’m not sure I understand how G1 learns to generate coherent text by just maximizing the probability of its samples.

In DelibGAN-2, the word-level scores from passing the first stage decoder’s output through the discriminator is used to score the sequence. This makes more sense to me w.r.t to why the first stage generator could learn to produce coherent text.

Review:
The writing could be clearer. Spacing is consistently an issue in w.r.t. to citations, referencing equations an figures, and using punctuations. Equations are difficult to understand, as well. See Equation 5. I’m confused by how the discriminator score of x_t is computed in equation 5. It seems to be the score of the multi-instance learner when applied to the concatenation of the previous generated tokens and the most recently generated token. This isn’t really a roll-out policy, however, since only a single step is taken in the future. It’s just scoring the next token according to the discriminator. In this case, I’m not sure what the summation over j is supposed to represent. It seems to index different histories when scoring word x_t. The authors should clarify exactly what the scoring procedure is for sequences generated by G2 and make sure that clarification matches what is written in the equation.

Because deliberation has previous been explored in Xia et al., 2017, the novelty of this work rests on the application of deliberation to GAN training of generative text models, as well as the development of the multi-instance discriminator for assigning word-level scores to the first-stage decoder's actions. However, it’s difficult to know how much of the improvement in performance is due to the modeling improvement because the evaluations are missing key details.

First, no information about the model’s hyperparameter setting were provided. Naturally, having two generators would double the number of parameters of the total model. Was an ablation run that looked at doubling the parameters of a single-generator? How powerful are the models being used in this new setup. With powerful models such as the OpenAI GPT being available for generative modeling, how powerful are the base units in this framework?

Second, I don’t have much intuition for how difficult the tasks being performed are. Because the authors aren’t evaluating on particularly difficult language datasets, they should provide statistics about the datasets w.r.t to vocabulary size, average length, etc.

Consequently, a lot of information that is necessary for evaluating the strengths and weaknesses of this paper are missing from the write-up.

Questions:
Equation 2 is slightly confusing. Is the representation of the word at that time step not conditioned on previous and future hidden states?

Is the multi-instance discriminator used in DelibGAN-II or is G2 only scored using the sentence level discriminator score?

Small:
-Wang & Wan, 2018a are definitely not the first work to discover that using word-level losses to train models for sentence-level evaluations is suboptimal
-Please fix issues with spacing. There are issues when citing papers, or referencing equations, or using punctuation.
-Equation 3 is missing a closing parenthesis on the softmax

---

### Official Review · AnonReviewer1 · 2018-11-04
**Limited novelty with scope limited to single sentence generation**

**Rating:** 4
**Confidence:** 4

**Review:**

This paper proposes using a deliberation network approach to text generation with GANs. It does this by using two decoders where the discriminator is trained on the second decoder and signals from that training are also used to improve the first decoder.

Since both approaches have been published before (deliberation networks and policy-gradient based GAN training), the novelty seems to be limited. Additionally, the experiments appear to mostly be focused on single sentence text generation, which is usually not an issue for maximum likelihood trained models. The poor samples from DelibMLE appear much worse than typical maximum likelihood trained language models. Since the model sizes and hyperparameters are not given, it's possible this is due to an overly small model. The lack of a detailed description of the model hyperparameters would also make this work very hard to reproduce.

There's also very little description of the datasets, for example, the length of a typical example. Finally, while the human evaluation results are interesting, the lack of standard deviations make it hard to measure the significance of the results, especially when only 100 sentences from each model is rated.

---

### Meta-Review · Area_Chair1 · 2018-12-13
**Reviewers agree upon rejection**

**Confidence:** 4
**Recommendation:** Reject

**Metareview:**

All reviewers are in agreement for a rejection decision.
Details below.